**Data Availability Statement:** All relevant data are within the paper.

**Funding:** The author(s) received no specific funding for this work.

# Prevalence and associated risk factors for mental health problems among patients with polycystic ovary syndrome in Bangladesh: A nationwide cross-sectional study

**Moynul Hasan[1], Sumaya Sultana[1], Md. Sohan📧[2], Shahnaj Parvin[1], Md. Ashrafur Rahman[3], Md. Jamal Hossain📧[4], Mohammad Saydur Rahman[1], Md. Rabiul Islam📧[2]***

**1** Department of Pharmacy, Jagannath University, Dhaka, Bangladesh, **2** Department of Pharmacy, University of Asia Pacific, Dhaka, Bangladesh, **3** Department of Pharmaceutical Sciences, North South University, Dhaka, Bangladesh, **4** Department of Pharmacy, State University of Bangladesh, Dhaka, Bangladesh

* robi.ayaan@gmail.com

## Abstract

### Background

Polycystic ovary syndrome (PCOS) is a common female reproductive endocrine problem worldwide. The prevalence of mental disorder is increasing among PCOS patients due to various physical, psychological, and social issues. Here we aimed to evaluate the mental health and associated factors among women suffering from PCOS in Bangladesh.

### Methods

We performed an online cross-sectional survey among 409 participants with PCOS using Google Forms. We used structured questionnaires to collect socio-demographic information and lifestyle-related factors. Also, we applied patient health questionnaire (PHQ-9), generalized anxiety disorder (GAD-7) scale, and UCLA loneliness (UCLA-3) scale for psychometric assessment of the participants. Finally, we applied several statistical tools and performed data interpretations to evaluate the prevalence of mental health disorders and associated factors among patients with PCOS in Bangladesh.

### Results

Prevalence of loneliness, generalized anxiety disorder and depressive illness among the women with PCOS were 71%, 88%, and 60%, respectively. Among the mental illness, mild, moderate, and severe cases were 39%, 18%, and 14% for loneliness; 39%, 23% and 26% for generalized anxiety disorder; and 35%, 18%, and 7% for depressive disorder. According to the present findings, obesity, financial condition, physical exercise, mealtime, food habit, daily water consumption, birth control method, and long-term oral contraceptive pills contribute to developing mental health disorders among females with PCOS in Bangladesh.

**Competing interests:** The authors have declared that no competing interests exist.

## Conclusion

According to present study results, high proportion of women suffering from PCOS experience several mental disorders in Bangladesh. Although several socio-demographic and lifestyle-related factors were found to be associated with the poor mental health of women with PCOS; however, PCOS itself is a condition that favors poor physical and psychological health. Therefore, we recommend proper treatment, public awareness, and a healthy lifestyle to promote the good mental health of women suffering from PCOS.

## Introduction

Polycystic ovary syndrome (PCOS) is a complex heterogeneous endocrine disorder related to the reproductive disease of females. In the female ovarian antral follicles, many cysts are seen to develop due to the improper balance of female sex hormones [1–4]. The prognosis of PCOS can be determined by examining three interconnected symptoms such as hyperandrogenism, ovulation disorders, or PCOM (polycystic ovary morphology) [5]. Hyperandrogenism can be identified by some visible signs including growing fatter, abdominal and subcutaneous fat, facial and body hair, hair loss, enlargement of the clitoris, deep voice, oily skin, acne, etc. [6]. One of the significant symptoms of PCOS is insulin resistance (IR). It is responsible for hyperinsulinemia leading to type II diabetes mellitus [7]. Another significant symptom of PCOS is sleep apnea, which occurs due to an imbalance in the sex steroid levels [8]. The etiology or genesis of this disease has not been fully known. There has been significant discussion concerning the genesis and pathological relationship of PCOS [9]. PCOS can be caused by a variety of reasons. PCOS can be caused by either genetics or a poor lifestyle, or a combination of both [1, 2]. According to several studies, a deficiency in insulin function may be the root cause of PCOS [10–12]. A dysfunction in the thyroid gland, CAH (congenital adrenal hyperplasia), elevated serum prolactin hormone, androgen-secreting tumors, and Cushing's syndrome are among the factors that might contribute to PCOS pathogenesis [2]. Depending on diagnostic criteria, it is found that PCOS is common among 21.27% of women [13]. In developed countries, the prevalence of PCOS is about 6–10% [14]. In Bangladesh, a significant percentage of women have been suffering from PCOS [15].

Women with PCOS have experienced adverse social, physical, emotional, and psychological consequences that negatively impacted their health-related quality of life [16]. As a result, their social and interpersonal relationships are hampered [17]. They experienced pessimistic, mocking behavior during treatment, which made them stressed and more discouraged. Even if they did not get enough support from their family members and they might be isolated from their families [18]. Therefore, they experience humiliation from society. Moreover, obesity, hirsutism, hair loss, menstrual abnormalities, and facial acne harmed their physical health. Furthermore, impaired infertility makes them humiliated and lowers their self-confidence [17]. Women with PCOS frequently report that they feel less attraction to their sexual life [19]. These reasons may justify a high divorce rate among women with PCOS [20]. Therefore, they are more prone to have psychological issues such as anxiety, depression, sadness, and loneliness than others [21, 22].

The prevalence of depression and anxiety symptoms is high among individuals with PCOS [23]. Several studies revealed that PCOS has a detrimental impact on the mental health of patients. Another study reported that 70–74% of PCOS patients were obese which indicates a

high level of dissatisfaction regarding their physical appearance [24]. A correlation has been found between physical obesity and depressive symptoms in the case of PCOS patients [25–27]. Also, anxiety is the most frequent psychological condition among women with PCOS [28, 29].

Anxiety can cause social isolation, lower quality of life, and increase the chance of developing another mental illness [29]. The development of anxiety symptoms in patients with PCOS is common [26]. Anxiety intensifies parallelly with the gradual development of hyperinsulinemia as well as hyperandrogenism, which are the typical symptoms of PCOS. Such parallel relation with anxiety is also found in the case of hormonal abnormalities of individuals with PCOS. It was found that women with PCOS who had a higher level of anxiety had more severe depression, indicating a possible relationship between these psychological disorders in PCOS women [29]. Thus, solid proof of a close relation between PCOS and psychological disorders is established.

There are several sex-specific factors involved in the poor mental health of women in Bangladesh [30–33]. Individuals with PCOS have to face many psycho-social problems in Bangladesh [34]. However, there are no studies in Bangladesh that adequately evaluated the mental health of women suffering from PCOS. Therefore, we aimed to assess the mental health of women suffering from PCOS in Bangladesh. Also, we tried to find the associated risk factors for the poor mental health of PCOS patients. We believe the present study findings will help to improve the quality of life of women suffering from PCOS.

## Materials and methods

### Study design and participants

We accomplished this cross-sectional online survey using Google Forms from June 15, 2021, to October 15, 2021. We consider the margin of error, response rate, and confidence level to be about 5%, 50%, and 95%, respectively. We required 384 responses to achieve 80% statistical power based on this distribution. Initially, we received responses from a total of 444 respondents. After sorting out all the answers, we ignored 35 respondents due to providing incomplete or partial information. After this step, we selected 409 respondents between 15 to 45 years of age. The objective of this study, eligibility requirement, and procedure of this study were known by all participants. At the initial stage of this survey, we obtained an electronic consent form from all participants. All respondents for the present study were of Bangladeshi nationality and lived in Bangladesh at the time of data collection. We included participants with a confirmed diagnosis of PCOS in this study. Participants over the age of 45 and under 18 were excluded from this study. The information was provided voluntarily by all participants.

### Estimations

We gathered information to find a correlation between the recent demographic profile and lifestyle-related factors with mental health problems among patients with PCOS. Later, with the help of mental health evaluation tools, we assessed the degrees of loneliness, depression, and generalized anxiety. We used a pre-structured questionnaire set developed by researchers to obtain the required information. At first, questionnaires were prepared in English and then converted into Bangla. To translate the questions, we took help from two Bangla native speakers (a medical graduate and a nonmedical individual) who have expertise in the English language. An independent author combined the translated versions to form a single Bangla forward version, and discrepancies were resolved with the assistance of a third author. A professional translator with experience in medical translation translated this Bangla version into English. An independent researcher combined these back-translated versions to create a single

English version [35]. Then, we conducted a preliminary test among the randomly selected small participants group to confirm the clarity and understanding of the questions. We excluded this pilot-testing information from the final study population data. Finally, we delivered the survey questionnaire in both Bangla and English to guarantee adequate comprehension. We collected information by sending the link to the questionnaire to participants via e-mail, Facebook, Messenger, WhatsApp as well as other social media platforms. We assisted with video conferences or phone conversations to resolve any issues or concerns regarding the questionnaire.

## Socio-demographic and biophysical measures

Most relevant socio-demographic information related to our study was collected from our respondents. We collected the most relevant sociodemographic information from the respondents. Collected data were regarding age, body mass index (BMI), marital status, education, economic impression, residence area, living status (with or without family), smoking habit, and family history of PCOS.

## Patient health questionnaire

The patient health questionnaire-9 (PHQ-9) is a globally recognized questionnaire having nine different questions to evaluate the depressive symptoms of respondents [36]. The total score for these self-administered questions ranges from 0 to 27 as the score allocated for each question is from 0 to 3 (0 = not at all; 1 = several days; 2 = more than a week; 3 = nearly every day). To express different levels of depression, the total score is divided into four distinct parts named as mild depression if the score is ≤9; 10–14 moderate depression if the score is 10–14; moderately severe depression if the score is 15–19 and severe depression if the score is ≥20 [37–41].

## Generalized anxiety disorder scale

The generalized anxiety disorder 7-item (GAD-7) scale is a valid and efficient tool for evaluating generalized anxiety symptoms. This scale has seven different questions for assessing symptoms if they persist for at least two weeks. Each question has four other scores ranging from 0 to 3 where 0 means "Not at all", 1 means "Several days", 2 means "more than half the days", and 3 means "Nearly every day". The total score for the GAD-7 ranges from 0 to 27, separated into four different segments indicating a different degree of anxiety (≤4 scores indicate no anxiety, 5–9 scores indicate mild anxiety, 10–14 scores indicate moderate anxiety and ≥15 scores indicate severe anxiety) [42].

## Loneliness scale

To calculate the degree of loneliness among respondents, we used UCLA Loneliness Scale (Version 3). This scale was created to make the response format easier to understand. It contains a total of 20 questions "How often do you feel" at the beginning of each question. This version of the UCLA Loneliness Scale includes 11 negatively phrased (lonely) and 9 positively phrased (non-lonely) questions. Here, the respondents had four options to respond: 'Never', 'Rarely', 'Sometimes', and 'Always' with scores 1,2,3, and 4, respectively. The score was reversed in the case of positively stated questions (i.e., 1 = 4, 2 = 3, 3 = 2, and 4 = 1). After then, the scores for each question are summed together; higher scores indicate greater degrees of loneliness [43].

## Data analysis

The data analysis was performed with the help of Microsoft Excel 2019 and Statistical Packages for Social Sciences (IBM SPSS) V.25.0. We utilized Microsoft Excel for data processing, sorting, coding, categorization, and tabulation. We next loaded the Excel file into the IBM SPSS program. Briefly, descriptive statistics were applied to analyze the characteristics of the study participants. To examine the differences between the group statistics, we performed the chi-square test. Binary logistic regression analysis was applied to find the risk ratios of having mental health problems due to socio-demographic and life-style related factors among participants. We evaluated statistically significant findings from analyses at $p < 0.05$.

## Ethics

The Research Ethics Committee of the Department of Pharmacy, Jagannath University, Dhaka, Bangladesh, reviewed and approved the study protocol (Ref: 003/2021). We conducted the present study following the principles stated in the Declaration of Helsinki. Also, we briefed the objective and purpose of this study at the beginning of the questionnaire. We obtained electronic consent from each patient before taking part in this study.

## Results

### Characteristics of respondents

The socio-demographic profile of the respondents was demonstrated in Table 1. Among 409 females, most of them (60.15%) were between 15–25 years of age, a significant proportion (36.67%) of them were between 26–35 years of age, and minor respondents (3.18%) were between 36–45 years of age. According to our analysis, we found that more than half (52.81%) of participants were married. Most of them were graduates (73.11%), low economic background women (60.39%), and non-smokers (95.84%). About 90.71% of women lived with their families, and around one-fifth (21.03%) of respondents were from rural areas.

### Lifestyle-related factors of respondents

We presented the lifestyle-related factors of respondents in Table 2. Among respondents, only 22.74% of respondents engaged in regular physical activity, with the majority of them exercising for less than 30 minutes (78%) on fewer than three days per week (78.97%). Among our participants, 24.45% of participants frequently had junk food, and most of them had mixed diet patterns (90.95%). On the questions of "having morning or evening snacks," "skipping any normal meals," and "eating meals on schedule," 7.83%, 40.5%, and 54.03% of respondents, respectively, provided a negative response. A minor proportion of participants consumed more than 8 glasses of water per day (21.03%), followed a birth control method (23.72%), and took oral contraceptive pills (12.96%).

### Psychometric parameters

Loneliness, generalized anxiety disorder, and depressive disorder were estimated to be 71%, 88%, and 60%, respectively (Fig 1). We estimated the prevalence of loneliness, generalized anxiety disorder, and depressive disorder in association with socio-demographic variables (Table 1). The high prevalence of loneliness was observed in (i) married versus unmarried (50.87% vs 49.13%, p = 0.221), (ii) individuals with BMI below 18.5 versus normal (82.61% vs 69.95%, p = 0.432), (iii) individuals with graduation or higher versus primary education background (73.70% vs 0.35%, p = 0.552), (iv) low versus high economic background (59.86% versus 13.84%, p = 0.682), (v) residing in urban versus rural area (79.58% vs 20.42%, p = 0.638),

**Table 1. Distribution of socio-demographic variables and their association with mental health problems among the patients suffering from the polycystic ovarian syndrome.**

| Socio-demographic parameters | Total (N = 409) | | Loneliness (N = 289) | | | | | Generalized anxiety disorder (N = 359) | | | | | Depressive disorder (N = 244) | | | | |
|---|---|---|---|---|---|---|---|---|---|---|---|---|---|---|---|---|---|
| | n | % | n | % | χ2 | df | p- value | n | % | χ2 | df | p- value | n | % | χ2 | df | p- value |
| Age in years | | | | | | | | | | | | | | | | | |
| 18–25 | 246 | 60.15 | 175 | 60.554 | 0.071 | 2 | 0.965 | 219 | 61.003 | 0.919 | 2 | 0.632 | 145 | 59.43 | 1.379 | 2 | 0.502 |
| 26–35 | 150 | 36.67 | 105 | 36.332 | | | | 129 | 35.933 | | | | 93 | 38.11 | | | |
| 36–45 | 13 | 3.18 | 9 | 3.114 | | | | 11 | 3.064 | | | | 6 | 2.46 | | | |
| Marital status | | | | | | | | | | | | | | | | | |
| Married | 216 | 52.81 | 147 | 50.87 | 1.498 | 1 | 0.221 | 188 | 52.37 | 0.232 | 1 | 0.630 | 124 | 50.82 | 0.963 | 1 | 0.326 |
| Unmarried | 193 | 47.19 | 142 | 49.13 | | | | 171 | 47.63 | | | | 120 | 49.18 | | | |
| BMI (kg/m2) | | | | | | | | | | | | | | | | | |
| Below 18.5 (CED) | 23 | 5.62 | 19 | 82.61 | 1.678 | 2 | 0.432 | 20 | 86.96 | 1.971 | 2 | 0.373 | 18 | 78.26 | 11.944 | 2 | **0.003** |
| 18.5–25 (normal) | 193 | 47.19 | 135 | 69.95 | | | | 165 | 85.49 | | | | 99 | 51.30 | | | |
| Above 25 (obese) | 193 | 47.19 | 135 | 69.95 | | | | 174 | 90.16 | | | | 127 | 65.80 | | | |
| Education | | | | | | | | | | | | | | | | | |
| Illiterate | 3 | 0.73 | 2 | 0.69 | 2.098 | 3 | 0.552 | 3 | 0.84 | 5.514 | 3 | 0.138 | 3 | 1.23 | 3.784 | 3 | 0.286 |
| Primary | 3 | 0.73 | 1 | 0.35 | | | | 2 | 0.56 | | | | 1 | 0.41 | | | |
| Secondary | 104 | 25.43 | 73 | 25.26 | | | | 97 | 27.02 | | | | 66 | 27.05 | | | |
| Graduate and above | 299 | 73.11 | 213 | 73.70 | | | | 257 | 71.58 | | | | 174 | 71.31 | | | |
| Profession | | | | | | | | | | | | | | | | | |
| Student | 200 | 48.89 | 143 | 49.48 | 1.273 | 3 | 0.736 | 175 | 48.75 | 3.322 | 3 | 0.345 | 120 | 49.18 | 1.981 | 3 | 0.576 |
| Service | 68 | 16.63 | 50 | 17.30 | | | | 56 | 15.59 | | | | 37 | 15.16 | | | |
| Business/Self-employed | 18 | 4.41 | 11 | 3.81 | | | | 17 | 4.74 | | | | 13 | 5.33 | | | |
| Unemployed | 123 | 30.07 | 85 | 29.41 | | | | 111 | 30.92 | | | | 74 | 30.33 | | | |
| Economic impression | | | | | | | | | | | | | | | | | |
| High | 59 | 14.43 | 40 | 13.84 | 0.765 | 2 | 0.682 | 47 | 13.09 | 6.098 | 2 | **0.047** | 38 | 15.57 | 1.741 | 2 | 0.419 |
| Middle | 103 | 25.18 | 76 | 26.30 | | | | 88 | 24.51 | | | | 65 | 26.64 | | | |
| Low | 247 | 60.39 | 173 | 59.86 | | | | 224 | 62.40 | | | | 141 | 57.79 | | | |
| Residence area | | | | | | | | | | | | | | | | | |
| Rural | 86 | 21.03 | 59 | 20.42 | 0.222 | 1 | 0.638 | 73 | 20.33 | 0.848 | 1 | 0.357 | 52 | 21.31 | 0.029 | 1 | 0.864 |
| Urban | 323 | 78.97 | 230 | 79.58 | | | | 286 | 79.67 | | | | 192 | 78.69 | | | |
| Living status | | | | | | | | | | | | | | | | | |
| With family | 371 | 90.71 | 263 | 91 | 0.101 | 1 | 0.750 | 329 | 91.64 | 3.042 | 1 | 0.081 | 218 | 89.34 | 1.337 | 1 | 0.248 |
| Without family | 38 | 9.29 | 26 | 9 | | | | 30 | 8.36 | | | | 26 | 10.66 | | | |
| Smoking habit | | | | | | | | | | | | | | | | | |
| Non-smoker | 392 | 95.84 | 277 | 95.85 | 0.000 | 1 | 0.995 | 343 | 95.54 | 0.665 | 1 | 0.415 | 232 | 95.08 | 0.881 | 1 | 0.348 |
| Smoker | 17 | 4.16 | 12 | 4.15 | | | | 16 | 4.46 | | | | 12 | 4.92 | | | |
| Family history of PCOS | | | | | | | | | | | | | | | | | |
| Yes | 74 | 18.09 | 47 | 16.26 | 2.226 | 1 | 0.136 | 68 | 18.94 | 1.427 | 1 | 0.232 | 50 | 20.49 | 2.349 | 1 | 0.125 |
| No | 335 | 81.91 | 242 | 83.74 | | | | 291 | 81.06 | | | | 194 | 79.51 | | | |

p-values are significant at 95% confidence interval (p < 0.05). Significant p-values are shown in bold. BMI, body mass index; CED, chronic energy deficiency; N, number.

(vi) staying with versus without family (91% versus 9%, p = 0.750), (vii) non-smoker versus smoker (95.85% vs 4.15%, p = 0.995). The frequency of having generalized anxiety disorder were higher in (i) married versus unmarried (52.37% vs 47.63%, p = 0.630), (ii) individuals

**Table 2. Distribution of lifestyle-related factors and their association with mental health problems among the patients suffering from polycystic ovarian syndrome.**

| Lifestyle-related factors | Total (N = 409) | | Loneliness (N = 289) | | | | | Generalized anxiety disorder (N = 359) | | | | | Depressive disorder (N = 244) | | | | |
|---|---|---|---|---|---|---|---|---|---|---|---|---|---|---|---|---|---|
| | n | % | n | % | χ2 | df | p-value | n | % | χ2 | df | p-value | n | % | χ2 | d f | p-value |
| Perform regular physical exercise | | | | | | | | | | | | | | | | | |
| Yes | 93 | 22.74 | 68 | 23.53 | 0.351 | 1 | 0.554 | 85 | 23.68 | 1.472 | 1 | 0.225 | 58 | 23.77 | 0.367 | 1 | 0.545 |
| No | 316 | 77.26 | 221 | 76.47 | | | | 274 | 76.32 | | | | 186 | 76.23 | | | |
| Physical exercise per day | | | | | | | | | | | | | | | | | |
| Less than 30 minutes | 319 | 78.00 | 226 | 78.20 | 0.024 | 1 | 0.876 | 279 | 77.72 | 0.133 | 1 | 0.715 | 201 | 82.38 | 6.76 | 1 | **0.009** |
| 30 minutes or more | 90 | 22.00 | 63 | 21.80 | | | | 80 | 22.28 | | | | 43 | 17.62 | | | |
| Physical exercise per week | | | | | | | | | | | | | | | | | |
| Less than 3 days | 323 | 78.97 | 229 | 79.24 | 0.042 | 1 | 0.838 | 279 | 77.72 | 2.795 | 1 | 0.095 | 193 | 79.1 | 0.006 | 1 | 0.940 |
| 3 or more days | 86 | 21.03 | 60 | 20.76 | | | | 80 | 22.28 | | | | 51 | 20.90 | | | |
| Frequency of taking junk food | | | | | | | | | | | | | | | | | |
| Occasionally | 121 | 29.59 | 83 | 28.72 | 5.218 | 3 | 0.157 | 104 | 28.97 | 1.537 | 3 | 0.674 | 73 | 29.92 | 12.9 | 3 | **0.005** |
| Once in a month | 45 | 11 | 26 | 8.99 | | | | 41 | 11.42 | | | | 32 | 13.11 | | | |
| Once in a week | 143 | 34.96 | 106 | 36.68 | | | | 128 | 35.65 | | | | 70 | 28.69 | | | |
| Frequently | 100 | 24.45 | 74 | 25.61 | | | | 86 | 23.96 | | | | 69 | 28.28 | | | |
| Diet pattern | | | | | | | | | | | | | | | | | |
| Vegetarian | 5 | 1.22 | 2 | 0.69 | 2.558 | 2 | 0.278 | 5 | 1.4 | 2.021 | 2 | 0.364 | 1 | 0.41 | 4.37 | 2 | 0.112 |
| Non-vegetarian | 32 | 7.83 | 24 | 8.31 | | | | 26 | 7.24 | | | | 22 | 9.02 | | | |
| Mixed | 372 | 90.95 | 263 | 91 | | | | 328 | 91.36 | | | | 221 | 90.57 | | | |
| Morning or afternoon snacks | | | | | | | | | | | | | | | | | |
| Always | 143 | 34.96 | 98 | 33.91 | 0.660 | 2 | 0.719 | 128 | 35.65 | 0.632 | 2 | 0.729 | 66 | 27.05 | 17.6 | 2 | **<0.001** |
| Sometimes | 234 | 57.21 | 167 | 57.79 | | | | 203 | 56.55 | | | | 154 | 63.11 | | | |
| Never | 32 | 7.83 | 24 | 8.30 | | | | 28 | 7.8 | | | | 24 | 9.84 | | | |
| Skip any regular meals | | | | | | | | | | | | | | | | | |
| Yes | 243 | 59.41 | 172 | 59.52 | 0.004 | 1 | 0.948 | 215 | 59.89 | 0.275 | 1 | 0.600 | 161 | 65.98 | 10.82 | 1 | **0.001** |
| No | 166 | 40.59 | 117 | 40.48 | | | | 144 | 40.11 | | | | 83 | 34.02 | | | |
| Taking meals on time | | | | | | | | | | | | | | | | | |
| Yes | 188 | 45.97 | 123 | 42.56 | 4.599 | 1 | **0.032** | 160 | 44.57 | 2.309 | 1 | 0.129 | 100 | 40.98 | 6.04 | 1 | **0.014** |
| No | 221 | 54.03 | 166 | 57.44 | | | | 199 | 55.43 | | | | 144 | 59.02 | | | |
| Daily water consumption | | | | | | | | | | | | | | | | | |
| Less than 8 glasses | 154 | 37.65 | 113 | 39.10 | 5.460 | 2 | 0.065 | 135 | 37.61 | 1.062 | 2 | 0.588 | 96 | 39.34 | 9.69 | 2 | **0.008** |
| 8 glasses | 169 | 41.32 | 124 | 42.91 | | | | 151 | 42.06 | | | | 87 | 35.66 | | | |
| More than 8 glasses | 86 | 21.03 | 52 | 17.99 | | | | 73 | 20.33 | | | | 61 | 25 | | | |
| Following any birth control method | | | | | | | | | | | | | | | | | |
| Yes | 97 | 23.72 | 69 | 23.88 | 0.014 | 1 | 0.907 | 90 | 25.07 | 2.973 | 1 | 0.085 | 47 | 19.26 | 6.632 | 1 | **0.010** |
| No | 312 | 76.28 | 220 | 76.12 | | | | 269 | 74.93 | | | | 197 | 80.74 | | | |
| Taking oral contraceptive pills | | | | | | | | | | | | | | | | | |
| Yes | 53 | 12.96 | 43 | 14.88 | 3.221 | 1 | 0.073 | 52 | 14.48 | 6.065 | 1 | **0.014** | 30 | 12.3 | 0.236 | 1 | 0.627 |
| No | 356 | 87.04 | 246 | 85.12 | | | | 307 | 85.52 | | | | 214 | 87.7 | | | |

p-values are significant at 95% confidence interval (p < 0.05). Significant p-values are shown in bold. N, number.

with graduation or higher versus primary education background (71.58% vs 0.56%, p = 0.138), (iii) low versus high economic background (62.40% versus 13.09%, p = 0.047), (iv) residing in urban versus rural area (79.67% vs 20.33%, p = 0.357), (v) staying with versus without family (91.64% versus 8.36%, p = 0.081), (vi) non-smoker versus smoker (95.54% vs 4.46%, p = 0.415). Similarly, respondents were more prone to have depressive disorder in (i) married

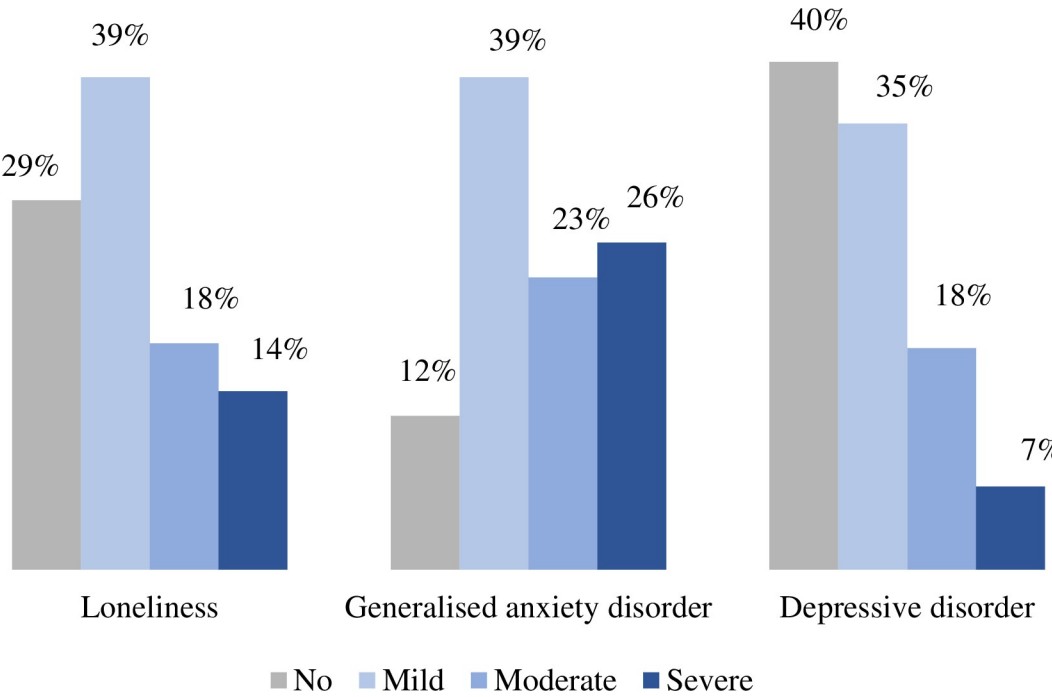

**Fig 1. Prevalence and distribution of mental health problems among patients with polycystic ovarian syndrome in Bangladesh.**

versus unmarried (50.82% vs 49.18%, p = 0.326), (ii) individuals with BMI below 18.5 versus normal (78.26% vs 51.30%, p = 0.003), (iii) individuals with graduation or higher versus primary education background (71.31% vs 0.41%, p = 0.286), (iv) low versus high economic background (57.79% versus 15.57%, p = 0.419), (v) residing in urban versus rural area (78.69% vs 21.31%, p = 0.864), (vi) staying with versus without family (89.34% versus 10.66%, p = 0.248), (vii) non-smoker versus smoker (95.08% vs 4.92%, p = 0.348).

At the same time, we estimated the prevalence of loneliness, generalized anxiety disorder, and depressive disorder associated with lifestyle-related factors (Table 2). The high prevalence of loneliness was observed in individuals (i) doing physical exercise irregularly versus regularly (76.47% vs 23.53%, p = 0.554), (ii) taking junk food once a week versus once in a month (36.68% vs 8.99%, p = 0.157), (iii) skipping regular meals versus not skipping regular meals (59.52% vs 40.48%, p = 0.948), (iv) taking meal not timely versus timely (57.44% vs 42.56%, p = 0.032), (v) consuming 8 glasses versus more than 8 glasses of water per day (42.91% vs 17.99%, p = 0.065), (iv) not taking versus taking oral contraceptive pills (85.12% vs 14.88%, p = 0.073). The frequency of having generalized anxiety disorder were higher in (i) doing physical exercise irregularly versus regularly (76.32% vs 23.68%, p = 0.225), (ii) doing exercise for less than 3 days versus at least 3 days (77.72% vs 22.28%, p = 0.095), (iii) skipping regular meals versus not skipping regular meals (59.89% vs 40.11%, p = 0.600), (iv) taking meal not timely versus timely (55.43% vs 44.57%, p = 0.129), (v) not following verses following any birth control method (74.93% vs 25.07%, p = 0.085), (vi) not taking versus taking oral contraceptive pills (85.52% vs 14.48%, p = 0.014). Similarly, respondents were more prone to have depressive disorder in individuals (i) doing physical exercise regularly for less than 30 minutes versus at least 30 minutes (82.38% versus 17.62%, p = 0.009), (ii) taking junk food occasionally versus once in a week (29.92% vs 28.69%, p = 0.005), (iii) skipping regular meals versus not skipping regular meals (65.98% vs 34.02%, p = 0.001), (iv) taking meal not timely versus timely (59.02%

vs 40.98%, p = 0.014), (v) consuming less than 8 glasses versus more than 8 glasses of water per day (39.34% vs 25%, p = 0.008), (vi) not following verses following any birth control method (80.74% vs 19.26%, p = 0.010).

## Regression analysis

We estimated the correlations between the independent and dependent variables of the socio-demographic profile with the help of binary regression analysis (Table 3). The chances of getting depressive disorder among PCOS patients were 0.293 times lower with individuals having BMI below 18.5 and 0.535 times lower with individuals having normal BMI than obese people (p = 0.003). Respondents living with family members were 2.97 times more likely to have a generalized anxiety disorder than respondents living away from family members (OR = 2.97, 95% CI 1.08–8.16, p = 0.034). The probability of having anxiety was 3.22 times higher in the high economic background participants than in the low financial background participants (OR = 3.22, 95% CI 1.28–8.13, p = 0.039). Similarly, we estimated the correlations between the independent and dependent variables of lifestyle-related factors with the help of binary regression analysis (Table 4).

## Discussion

The most frequent ovarian condition related to high androgen levels in women is PCOS [44]. Chronic anovulation, hyperandrogenism, and metabolic abnormalities are all prevalent symptoms in women with PCOS. Women with PCOS are more likely to be obese, have IR, and hence are predisposed to glucose intolerance [45]. According to some recent studies, hyperandrogenemia (HA) and IR have been demonstrated as the core etiology and prime endocrine features of PCOS. The major causes of PCOS are HA and IR, and they can interact with one another in the onset and progression of the disease [46]. Women with PCOS have experienced multiple physical abnormalities, including obesity, acne, and hirsutism, along with prevalent menstrual disorders. This condition includes several psychosocial aspects in addition to numerous physical difficulties [47]. Various psychiatric disorders include depression, generalized anxiety disorder, personality disorders, social phobia, obsessive-compulsive disorder (OCD), attention deficit hyperactivity disorder (ADHD), bipolar affective disorder, schizophrenia, and eating disorders. Depression and generalized anxiety disorder are the highest prevalence of psychiatric disorders among patients with PCOS [48]. The present study was conducted to investigate the mental health problems such as loneliness, generalized anxiety disorder, and depressive disorder among women suffering from PCOS. According to our analysis, from the perspective of the socio-demographic profile, we concluded that marital status, education, financial background, area of residence, smoking habit, and family history of PCOS might be responsible for developing mental health issues among our participants. Therefore, an earlier study suggested that lifestyle change can be an option for the effective management of PCOS [49]. Another study revealed environmental and lifestyle factors contributing poor mental health of patients with PCOS [50]. An absolute correlation could have been established between educational qualifications and loneliness, anxiety, and depression. The respondents who had done graduation or held a degree above graduation complained of experiencing more loneliness, anxiety, and depression than the other participants having educational qualifications other than graduation or above. Financial conditions may have a direct effect on mental health issues. Participants with low economic impressions were going through more mental health problems. Most of our respondents lived with their families, and almost all of them complained of experiencing such mental health problems. From the perspective of lifestyle-related factors, we concluded that physical exercise, duration of physical exercise, food

**Table 3. Regression analysis of socio-demographic variables and their association with mental health problems among the patients suffering from polycystic ovarian syndrome.**

| Socio-demographic parameters | Loneliness (N = 289) | | | Generalized anxiety disorder (N = 359) | | | Depressive disorder (N = 244) | | |
|---|---|---|---|---|---|---|---|---|---|
| | OR | 95% CI | p-value | OR | 95% CI | p-value | OR | 95% CI | p-value |
| Age in years | | | | | | | | | |
| 15–25 | 0.828 | 0.176–3.888 | 0.963 | 0.387 | 0.031–4.816 | 0.517 | 1.843 | 0.406–8.369 | 0.669 |
| 26–35 | 0.890 | 0.207–3.822 | | 0.294 | 0.029–3.028 | | 1.921 | 0.460–8.024 | |
| 36–45 | 1 | | | 1 | | | 1 | | |
| Marital status | | | | | | | | | |
| Married | 1.511 | 0.777–2.939 | 0.224 | 1.956 | 0.753–5.082 | 0.168 | 1.329 | 0.701–2.519 | 0.384 |
| | 1 | | | 1 | | | 1 | | |
| BMI (kg/m2) | | | | | | | | | |
| Below 18.5 (CED) | 0.490 | 0.160–1.504 | 0.446 | 0.884 | 0.246–3.173 | 0.377 | 0.293 | 0.104–0.820 | 0.003 |
| 18.5–25 (normal) | 0.490 | | | 1.374 | 0.373–5.054 | | 0.535 | 0.190–1.504 | |
| Above 25 (obese) | 1 | | | 1 | | | 1 | | |
| Education | | | | | | | | | |
| Illiterate | 1.230 | 0.707–2.139 | 0.681 | 0.732 | 0.289–1.856 | 0.270 | 0.764 | 0.444–1.318 | 0.794 |
| Primary | 0.777 | 0.053–11.290 | | 0.815 | 0.537–1.365 | | 0.126 | 0.105–0.373 | |
| Secondary | 0.289 | 0.016–5.201 | | 0.022 | 0.000–1.018 | | 1.286 | 0.074–22.376 | |
| Graduate and above | 1 | | | 1 | | | 1 | | |
| Profession | | | | | | | | | |
| Student | 0.577 | 0.181–1.842 | 0.584 | 2.307 | 0.212–25.116 | 0.334 | 1.053 | 0.307–3.619 | 0.194 |
| Service | 1.221 | 0.569–2.620 | | 0.714 | 0.246–2.074 | | 0.470 | 0.226–0.979 | |
| Business/Self-employed | 0.801 | 0.389–1.651 | | 0.411 | 0.133–1.272 | | 0.708 | 0.345–1.452 | |
| Unemployed | 1 | | | 1 | | | 1 | | |
| Economic impression | | | | | | | | | |
| High | 1.162 | 0.585–2.306 | 0.841 | 3.227 | 1.281–8.131 | **0.039** | 0.767 | 0.382–1.542 | 0.362 |
| Middle | 1.257 | 0.587–2.692 | | 1.840 | 0.702–4.824 | | 1.114 | 0.516–2.409 | |
| Low | 1 | | | 1 | | | 1 | | |
| Residence area | | | | | | | | | |
| Rural | 0.939 | 0.533–1.653 | 0.827 | 0.536 | 0.246–1.165 | 0.115 | 1.182 | 0.673–2.075 | 0.561 |
| Urban | 1 | | | 1 | | | 1 | | |
| Living status | | | | | | | | | |
| With family | 1.391 | 0.626–3.093 | 0.418 | 2.976 | 1.085–8.164 | **0.034** | 0.789 | 0.342–1.822 | 0.579 |
| Without family | 1 | | | 1 | | | 1 | | |
| Smoking habit | | | | | | | | | |
| Non-smoker | 0.746 | 0.224–2.488 | 0.633 | 0.312 | 0.029–3.309 | 0.334 | 0.766 | 0.221–2.660 | 0.675 |
| Smoker | 1 | | | 1 | | | 1 | | |
| Family history of PCOS | | | | | | | | | |
| Yes | 1.553 | 0.873–2.762 | 0.134 | 0.510 | 0.194–1.340 | 0.172 | 0.866 | 0.479–1.566 | 0.635 |
| No | 1 | | | 1 | | | 1 | | |

p-values are significant at 95% confidence interval (p < 0.05). Significant p-values are shown in bold. BMI, body mass index; CED, chronic energy deficiency; N, number.

habits, daily water consumption, birth control method, and the use of contraceptive pills might be responsible for developing mental health issues among our respondents. The women who performed physical exercise at least 30 minutes a day for at least 3 days a week had experienced better mental health than other participants. We found that a good food habit, not taking any morning and evening snacks, more than 8 glasses of water consumption per day, all of

**Table 4. Regression analysis of lifestyle-related factors and their association with mental health problems among the patients suffering from the polycystic ovarian syndrome.**

| Lifestyle-related factors | Loneliness (N = 289) | | | | Generalized anxiety disorder (N = 359) | | | | Depressive disorder (N = 244) | | | |
|---|---|---|---|---|---|---|---|---|---|---|---|---|
| | OR | df | 95% CI | p-value | OR | df | 95% CI | p-value | OR | df | 95% CI | p-value |
| Perform regular physical | | | | | | | | | | | | |
| Yes | 0.535 | 1 | 0.223–1.286 | 0.162 | 0.717 | 1 | 0.193–2.662 | 0.619 | 0.552 | 1 | 0.237–1.286 | 0.169 |
| No | 1 | | | | 1 | | | | 1 | | | |
| Physical exercise per day | | | | | | | | | | | | |
| Less than 30 minutes | 1.029 | 1 | 0.574–1.843 | 0.925 | 1.062 | 1 | 0.440–2.562 | 0.893 | 0.60 | 1 | 0.345–1.048 | 0.073 |
| 30 minutes or more | 1 | | | | 1 | | | | 1 | | | |
| Physical exercise per week | | | | | | | | | | | | |
| Less than 3 days | 0.697 | 1 | 0293–1.655 | 0.413 | 2.264 | 1 | 0.549–9.343 | 0.259 | 0.660 | 1 | 0.282–1.545 | 0.338 |
| 3 or more days | 1 | | | | 1 | | | | 1 | | | |
| Frequency of taking junk food | | | | | | | | | | | | |
| Occasionally | 1.168 | 2 | 0.641–2.127 | 0.615 | 0.653 | 2 | 0.278–1.530 | 0.591 | 1.95 | 2 | 1.082–3.527 | 0.078 |
| Once in a month | 0.846 | | 0.494–1.448 | | 0.760 | | 0.344–1.678 | | 1.100 | | 0.651–1.859 | |
| Once in a week | 1 | | | | 1 | | | | 1 | | | |
| Frequently | 1 | | | | 1 | | | | 1 | | | |
| Diet pattern | | | | | | | | | | | | |
| Vegetarian | 2.985 | 2 | 0.388–22.945 | 0.537 | 0.000 | 2 | 0.000 | 0.449 | 11.4 | 2 | 0.946–137.669 | 0.158 |
| Non-vegetarian | 3.580 | | 0.377–33.965 | | 0.000 | | 0.000 | | 10.86 | | 0.766–154.234 | |
| Mixed | 1 | | | | 1 | | | | 1 | | | |
| Morning or afternoon snacks | | | | | | | | | | | | |
| Always | 0.759 | 2 | 0.452–1.276 | 0.446 | 1.304 | 2 | 0.604–2.814 | 0.733 | 0.52 | 2 | 0.320–0.858 | 0.007 |
| Sometimes | 1.302 | | 0.507–3.339 | | 1.376 | | 0.374–5.056 | | 2.004 | | 0.754–5.326 | |
| Never | 1 | | | | 1 | | | | 1 | | | |
| Skip any regular meals | | | | | | | | | | | | |
| Yes | 1.022 | 1 | 0.624–1.674 | 0.932 | 0.682 | 1 | 0.331–1.404 | 0.299 | 0.642 | 1 | 0.402–1.025 | 0.064 |
| No | 1 | | | | 1 | | | | 1 | | | |
| Taking meals on time | | | | | | | | | | | | |
| Yes | 1.631 | 1 | 1.012–2.626 | **0.044** | 1.890 | 1 | 0.943–3.787 | 0.073 | 1.18 | 1 | 0.746–1.881 | 0.473 |
| No | 1 | | | | 1 | | | | 1 | | | |
| Daily water consumption | | | | | | | | | | | | |
| Less than 8 glasses | 1.861 | 2 | 1.016–3.409 | 0.054 | 1.758 | 2 | 0.740–4.174 | 0.334 | 0.41 | 2 | 0.222–0.786 | 0.026 |
| 8 glasses | 2.078 | | 1.110–3.890 | | 1.855 | | 0.763–4.507 | | 0.54 | | 0.285–1.053 | |
| More than 8 glasses | 1 | | | | 1 | | | | 1 | | | |
| Following any birth control | | | | | | | | | | | | |
| Yes | 1.087 | 1 | 0.560–2.111 | 0.805 | 0.652 | 1 | 0.234–1.816 | 0.413 | 1.737 | 1 | 0.903–3.343 | 0.098 |
| No | 1 | | | | 1 | | | | 1 | | | |
| Taking oral contraceptive pills | | | | | | | | | | | | |
| Yes | 0.405 | 1 | 0.169–0.970 | **0.043** | 0.117 | 1 | 0.013–1.056 | 0.056 | 0.662 | 1 | 0.299–1.466 | 0.310 |
| No | 1 | | | | 1 | | | | 1 | | | |

p-values are significant at 95% confidence interval (p < 0.05). Significant p-values are shown in bold. N, number.

these lifestyles may contribute to achieving great mental health among PCOS women. We also found that women who were not taking contraceptive pills and not following any birth control methods had some mental health issues.

Few studies have been carried out to assess the mental health problems of women having PCOS. In a study of psychiatric disorders in women with PCOS, women with this condition

were shown to have more significant risks for depression and anxiety disorders than the general population [51]. Patients with PCOS have higher mental stress due to clinical manifestations of the menstrual problem, and as a result, patients with PCOS require more extra therapies for their symptoms than others. This extra mental pressure could lead to depressive, and anxiety disorders [52]. It has been suggested by various studies that PCOS has an impaired sexual function that may be responsible for the poorer mental health of these women with a higher prevalence of anxiety and depression [53, 54]. According to one study, generalized anxiety disorder and depressive disorder symptoms in women with PCOS are more closely associated with obesity [55]. Another study found that PCOS women with BMI greater than 30 have higher depression levels than women with lower BMI [53]. Despite this, our research found not much influence of higher BMI or obesity on depressive symptoms or signs of generalized anxiety disorder in women with PCOS.

To the best of our knowledge, our study is the first approach in Bangladesh to assess the mental health problem of PCOS women from the perspective of socio-demographic profile and lifestyle-related factors simultaneously. We found some key factors of the socio-demographic profile, including education, economic impression, marital status, and family history of PCOS, as well as key factors of lifestyle-related factors including the total duration of physical exercise, food habits, daily water consumption that may contribute to the development of loneliness, generalized anxiety disorder, and depressive disorder. As the etiology of the PCOS is not fully known, the socio-demographic profile of respondents may not be a changeable factor, and leading to healthy lifestyle aid in mental health [56], the adaptation of a good lifestyle may help the PCOS women to maintain a life with better mental health. As per our analysis, 71%, 88%, and 60% of respondents are struggling with loneliness, anxiety, and depression, respectively, and the socio-demographic profile and lifestyle-related factors don't have a direct relationship with such a mental disorder. One earlier meta-analysis reported that the prevalence of depression and anxiety were 36.6% and 41.9% among women suffering from PCOS [57]. It can be concluded that PCOS and its consequences are responsible for increased mental health problems of women suffering from this hormonal disease [58, 59]. The PCOS cases are increasing in Bangladesh over the last decades due to the changed lifestyle and some other demographic factors. However, we do not have actual data regarding the mental health problems due to PCOS. This study's findings would help the healthcare authorities and allied healthcare professionals in designing the interventional approach to support the women suffering from PCOS. Also, the findings might help to create awareness among the general population about the consequences of PCOS and its preventive measures.

## Potential limitations of the study

The shortcomings of this study should be discussed. The use of Google forms for self-reporting surveys may introduce bias. Furthermore, this survey does not include those who do not have an internet connection. The inclusion of offline participants or offline surveys or interviews might have provided a different estimate of our hypothesis. Furthermore, we are unable to examine these mental health concerns over time in our cross-sectional investigation. PCOS was clinically diagnosed by physicians initially a few times back and was not assessed again during or before our data collection. On the other hand, PCOS is thought to be a permanent disorder. Furthermore, due to the cross-sectional nature of this investigation, we were unable to determine causation.

## Practical implications

The current study findings have widespread implications. The association of loneliness, generalized anxiety disorder and depressive disorder with PCOS women in Bangladesh is proven.

One can easily differentiate the key socio-demographic and lifestyle-related factors that may have contributed to the development of such mental health issues. The healthcare authority should pay attention to the development of guidelines that may converge the physical and psychological health of women with PCOS while physicians give proper care to them. At the same time, this study can contribute to thinking about the mental health of women struggling with PCOS worldwide.

## Conclusion

The present study results suggest that a high proportion of women suffering from PCOS have several mental health problems. This additional mental health burden among PCOS patients is due to demographics, physical health, lifestyle factors, psychological issues, and social reasons. Therefore, women suffering from PCOS have high a tendency to develop mental health problems. Based on the present findings, we recommend proper therapeutic interventions, public awareness, and a healthy lifestyle to promote the good mental health of women suffering from PCOS. However, further studies investigating the degree of loneliness, anxiety, and depression among women with PCOS to know the actual gravity of the issue.

## Acknowledgments

We thank all the participants for their cooperation to this study.

## Author Contributions

**Conceptualization:** Moynul Hasan, Sumaya Sultana, Mohammad Saydur Rahman, Md. Rabiul Islam.

**Data curation:** Sumaya Sultana, Md. Sohan, Shahnaj Parvin, Md. Ashrafur Rahman, Md. Jamal Hossain.

**Formal analysis:** Md. Ashrafur Rahman, Md. Jamal Hossain, Mohammad Saydur Rahman, Md. Rabiul Islam.

**Investigation:** Md. Ashrafur Rahman, Md. Jamal Hossain.

**Methodology:** Moynul Hasan, Mohammad Saydur Rahman, Md. Rabiul Islam.

**Project administration:** Moynul Hasan, Md. Rabiul Islam.

**Supervision:** Moynul Hasan, Md. Rabiul Islam.

**Writing – original draft:** Sumaya Sultana, Md. Sohan, Shahnaj Parvin.

**Writing – review & editing:** Moynul Hasan, Mohammad Saydur Rahman, Md. Rabiul Islam.

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
