## [Decision Letter · Decision Letter 0]

19 Apr 2022

PONE-D-21-40627Prevalence and associated risk factors for mental health problems among patients with polycystic ovary syndrome in Bangladesh: A nationwide cross- sectional studyPLOS ONE

Dear Dr. Islam,

Thank you for submitting your manuscript to PLOS ONE. After careful consideration, we feel that it has merit but does not fully meet PLOS ONE’s publication criteria as it currently stands. Therefore, we invite you to submit a revised version of the manuscript that addresses the points raised during the review process.

Two reviewers addressed several major concerns about your manuscript. Please revise your manuscript according to reviewer's comments.

We look forward to receiving your revised manuscript.

Kind regards,

Kenji Hashimoto, PhD

Academic Editor

PLOS ONE

Journal Requirements:

Reviewers' comments:

Reviewer's Responses to Questions

**Comments to the Author**

1. Is the manuscript technically sound, and do the data support the conclusions?

Reviewer #1: Yes

Reviewer #2: Yes

2. Has the statistical analysis been performed appropriately and rigorously? 

Reviewer #1: Yes

Reviewer #2: Yes

3. Have the authors made all data underlying the findings in their manuscript fully available?

Reviewer #1: Yes

Reviewer #2: No

4. Is the manuscript presented in an intelligible fashion and written in standard English?

Reviewer #1: No

Reviewer #2: Yes

5. Review Comments to the Author

Reviewer #1: I reviewed the manuscript entitled 'Prevalence and associated risk factors for mental health problems among patients withpolycystic ovary syndrome in Bangladesh: A nationwide cross-sectional study' and definitely this has merits in terms of women's health care.However, the following points have to be considered very seriously regarding the manuscript.1. In the abstract, focus on the method and result in more details. Do not put more words in the introduction. Secondly, the conclusion should be concise and must be a very accurate reflection of the study 2. The introduction section is written like a literature review. Reduce a minimum of 35 % of the unnecessary content. This would also help you in reducing a number of references.3. The aim of the study is very superficially presented. It should be very clear and concise.4. In the study design, what is the meaning od expected to have. It means you have not considered it yet.5. The mentioned sentence 'Furthermore,when we gathered the data, all respondents were of Bangladeshi nationality and lived inBangladesh' is one of the example of poor language editing. Revise whole manuscript for English grammar and language.6. What about the validation of the questionnaire.7. The details mentioned in DATA COLLECTION SUBHEADING are really annoying to the readers.8. Remove the unnecessary sentences or repeating sentences form the method section. make it more presented scientifically.9. For one method or instrument used, put one valid reference only. I do not understand why there is too many references are used in Generalized Anxiety Disorder Scale and Loneliness Scale.10. We also utilized bar graphs to compare the distribution of mental health problemsaccording to the severity.---Not needed here to write.11. , HA (hyperandrogenemia) and IR (insulin resistance)- First write full form then abbreviations.12. Remove this-----However, this study contains several noteworthy findings. To begin, the current study assessedthe three primary psychological difficulties associated with women having PCOS. As we usedGoogle forms for data collection, which allowed for the collecting of data from people of allsocioeconomic backgrounds and educational levels in a timely manner. Moreover, we employedthe mental health assessment scales in Bangla, which ensured that the questions were clearlyunderstood.---from the manuscript.13. Conclusion looks lengthy. I recommend making it concise.14. All the references must mention the DOI.

Reviewer #2: This is a valuable study about the mental health problem of PCOS women from the perspective of socio-demographic profile and lifestyle-related factors in Bangladesh.

However, there are some comments to the author.

1. What is the novelty of this study? Is it only that it is the first study in Bangladesh? The author should clearly state the novelty.

2. What is the message in Figure 1? The author should provide a discussion comparing the results with previous reports.

3. In Results, the description “Regression analysisのRespondents who took morning or afternoon snacks sometimes were more likely to have the depressive disorder than the respondents who never took snacks (OR=2.00, 95% CI 0.75-5.32, p=0.007)” is inappropriate because the 95% CI straddles 1.

4. In Discussion, there is the description “According to our analysis, from the perspective of the socio-demographic profile, we concluded that marital status, education, financial background, area of residence, smoking habit, and family history of PCOS might be responsible for developing mental health issues among our participants.” More consideration should be given to each of these factors. How does this compare to previous reports? The background of significant differences in each factor should be examined in detail.

5. In Discussion, repeated mention of the result is not required. “About 59.86%, 62.40%, and 57.79% of participants with low economic backgrounds were suffering from loneliness, anxiety, and depressive disorder. Almost half of the women among our participants have complained about their married life which was responsible for their poor mental health. According to our findings, women living in urban area (loneliness=79.58%, generalized anxiety disorder=78.67%, depressive disorder=78.69%), having no smoking habit (loneliness=95.85%, generalized anxiety disorder=95.54%, depressive disorder=95.08%), and having no family history of PCOS (loneliness=53.74%, generalized anxiety disorder=81.06%, depressive disorder=79.51%) might be more at risk of mental health problems.”

6. In Discussion, what is the meaning of a, b and c in the description “As per our analysis, the a%, b%, and c% of respondents are struggling with loneliness, anxiety, and depression, respectively”?

6. PLOS authors have the option to publish the peer review history of their article (what does this mean?). If published, this will include your full peer review and any attached files.

Reviewer #1: **Yes: **Dr. Mumammad Sayeed Akhtar

Reviewer #2: No

---

## [Author Response · Author response to Decision Letter 0]

23 May 2022

Dear Editors and Reviewers,

Thank you for your letter and the reviewers' comments on our manuscript entitled "Prevalence and associated risk factors for mental health problems among patients with polycystic ovary syndrome in Bangladesh: A nationwide cross- sectional study" (Manuscript ID PONE-D-21-40627). All the comments were valuable and helpful to the revision and improvement of the manuscript. We have carefully studied the comments and made corrections, which we hope will merit your approval. We marked the revised portions using track changes. Our point-by-point answers to the reviewers’ comments appear at the end of this letter.

We earnestly appreciate the Editors'/Reviewers' work. We hope that after this revision, the paper will be deemed fit for publication. We would be glad to respond to any further questions and comments that you may have. 

Once again, thank you very much for your comments and suggestions.

Best regards,

Md. Rabiul Islam, PhD

Assistant Professor, Department of Pharmacy, University of Asia Pacific, 74/A Green Road, Farmgate, Dhaka-1205, Bangladesh. Email: robi.ayaan@gmail.com; Cell: +8801916031831

Point by point authors’ responses to the reviewers

Manuscript ID PONE-D-21-40627

Title: Prevalence and associated risk factors for mental health problems among patients with polycystic ovary syndrome in Bangladesh: A nationwide cross- sectional study

Reviewer #1

I reviewed the manuscript entitled 'Prevalence and associated risk factors for mental health problems among patients with polycystic ovary syndrome in Bangladesh: A nationwide cross-sectional study' and definitely this has merits in terms of women's health care. However, the following points have to be considered very seriously regarding the manuscript.

Author’s response

Thank you for your review and encouraging comments on our manuscript. We have addressed all your observation very carefully in our revised manuscript. 

Comment 1. In the abstract, focus on the method and result in more details. Do not put more words in the introduction. Secondly, the conclusion should be concise and must be a very accurate reflection of the study 

Author’s response

Thank you for your observation. We have now revised the abstract following your suggestion (Page 2, line 5-34; page 3, line 1-2). 

Comment 2. The introduction section is written like a literature review. Reduce a minimum of 35 % of the unnecessary content. This would also help you in reducing a number of references.

Author’s response

Thank you for your suggestion. We have now reduced the unnecessary contents from the introduction part of our manuscript. 

Comment 3. The aim of the study is very superficially presented. It should be very clear and concise. 

Author’s response

Thank you again for your valuable suggestion. We have now written the aim of this study very clearly and concisely in the revised manuscript (page 5, line 16-20). 

Comment 4. In the study design, what is the meaning od expected to have. It means you have not considered it yet. 

Author’s response

Thank you for your opinion. We have corrected this accordingly (page 5, line 26; page 6, line 4). 

Comment 5. The mentioned sentence 'Furthermore, when we gathered the data, all respondents were of Bangladeshi nationality and lived in Bangladesh' is one of the example of poor language editing. Revise whole manuscript for English grammar and language.

Author’s response

Thank you for your comments and suggestions. According to your advice, the whole manuscript has been edited by a person who is proficient in written English. We hope that after this language edit, the paper will be considered suitable for publication.

Comment 6. What about the validation of the questionnaire.

Author’s response

Thank you for your observation. The questionnaires/scales used in this study are not validated. However, the questionnaires/scales were pilot-tested and we discussed this information at the method section (page 6, line 20-24). 

Comment 7. The details mentioned in DATA COLLECTION SUBHEADING are really annoying to the readers.

Author’s response

Thank you for your opinion. We have now deleted Data Collection subheading and merged the relevant text with previous section (page 6, line 25). 

Comment 8. Remove the unnecessary sentences or repeating sentences form the method section. make it more presented scientifically.

Author’s response

Thank you for your valuable observation. We edited the whole method section and deleted unnecessary and repeating sentences. 

Comment 9. For one method or instrument used, put one valid reference only. I do not understand why there is too many references are used in Generalized Anxiety Disorder Scale and Loneliness Scale.

Author’s response

Thank you for your suggestion. We have now mentioned one valid reference for Generalized Anxiety Disorder Scale and another one for Loneliness Scale (page 7, line 28, ref. 42; page 8, line 6, ref. 43). 

Comment 10. We also utilized bar graphs to compare the distribution of mental health problems according to the severity. ---Not needed here to write.

Author’s response

Thank you for your opinion. We have deleted this information from revised version (page 8 line 16-17). 

Comment 11. HA (hyperandrogenemia) and IR (insulin resistance)- First write full form then abbreviations.

Author’s response

Thank you for your observation. We have corrected this. 

Comment 12. Remove this-----However, this study contains several noteworthy findings. To begin, the current study assessed the three primary psychological difficulties associated with women having PCOS. As we used Google forms for data collection, which allowed for the collecting of data from people of all socioeconomic backgrounds and educational levels in a timely manner. Moreover, we employed the mental health assessment scales in Bangla, which ensured that the questions were clearly understood. ---from the manuscript.

Author’s response

Thank you for your suggestion. We have deleted this portion for the revised manuscript (page 14, line 31-32; page 15, line 1-4.). 

Comment 13. Conclusion looks lengthy. I recommend making it concise.

Author’s response

Thank you for your observation. We present a concise conclusion in the revised manuscript. 

Comment 14. All the references must mention the DOI

Author responses

Thank you for your suggestion. We added DOI number/link with all references. 

Reviewer #2

This is a valuable study about the mental health problem of PCOS women from the perspective of socio-demographic profile and lifestyle-related factors in Bangladesh.

However, there are some comments to the author.

Author’s response

Thank you for your review and appreciation. We have made all necessary revisions following your suggestion. 

Comment 1. What is the novelty of this study? Is it only that it is the first study in Bangladesh? The author should clearly state the novelty.

Author’s response

Thank you for your observation. The PCOS cases in increasing in Bangladesh over last decades due to the changed lifestyle and some other demographic factors. However, we do not have actual data regarding the mental health problems due to PCOS. This study findings would help the healthcare authorities and allied healthcare professionals to design the interventional approach to support the women suffering from PCOS. Also, the findings might help to create awareness among the general population about the consequences of PCOS and preventives measures of it (page 14, line 14-19).

Comment 2. What is the message in Figure 1? The author should provide a discussion comparing the results with previous reports.

Author’s response

Thank you for your suggestion. We have now compared our findings with previous reports in the revised manuscript (page 14, line 7-11). 

Comment 3. In Results, the description “Regression analysisのRespondents who took morning or afternoon snacks sometimes were more likely to have the depressive disorder than the respondents who never took snacks (OR=2.00, 95% CI 0.75-5.32, p=0.007)” is inappropriate because the 95% CI straddles 1.

Author’s response

Thank you for your observation. We eliminated this portion from result section (page 11, line 19-21). 

Comment 4. In Discussion, there is the description “According to our analysis, from the perspective of the socio-demographic profile, we concluded that marital status, education, financial background, area of residence, smoking habit, and family history of PCOS might be responsible for developing mental health issues among our participants.” More consideration should be given to each of these factors. How does this compare to previous reports? The background of significant differences in each factor should be examined in detail.

Author’s response

Thank you for your valuable suggestion. We have now added some information from earlier studies to compare these factors with present findings (page 12, line 19-21). 

Comment 5. In Discussion, repeated mention of the result is not required. “About 59.86%, 62.40%, and 57.79% of participants with low economic backgrounds were suffering from loneliness, anxiety, and depressive disorder. Almost half of the women among our participants have complained about their married life which was responsible for their poor mental health. According to our findings, women living in urban area (loneliness=79.58%, generalized anxiety disorder=78.67%, depressive disorder=78.69%), having no smoking habit (loneliness=95.85%, generalized anxiety disorder=95.54%, depressive disorder=95.08%), and having no family history of PCOS (loneliness=53.74%, generalized anxiety disorder=81.06%, depressive disorder=79.51%) might be more at risk of mental health problems.”

Author’s response

Thank you for your valuable suggestion. We omitted this portion from our revised manuscript (page 12, line 27-32; page 13, line 1-3). 

Comment 6. In Discussion, what is the meaning of a, b and c in the description “As per our analysis, the a%, b%, and c% of respondents are struggling with loneliness, anxiety, and depression, respectively”?

Author’s response

Thank you for your observation. We have corrected this information in the revised manuscript (page 14, line 7).

---

## [Decision Letter · Decision Letter 1]

6 Jun 2022

Prevalence and associated risk factors for mental health problems among patients with polycystic ovary syndrome in Bangladesh: A nationwide cross- sectional study

PONE-D-21-40627R1

Dear Dr. Islam,

We’re pleased to inform you that your manuscript has been judged scientifically suitable for publication and will be formally accepted for publication once it meets all outstanding technical requirements.

Kind regards,

Kenji Hashimoto, PhD

Section Editor

PLOS ONE

Additional Editor Comments (optional):

Reviewers' comments:

Reviewer's Responses to Questions

**Comments to the Author**

1. If the authors have adequately addressed your comments raised in a previous round of review and you feel that this manuscript is now acceptable for publication, you may indicate that here to bypass the “Comments to the Author” section, enter your conflict of interest statement in the “Confidential to Editor” section, and submit your "Accept" recommendation.

Reviewer #1: All comments have been addressed

Reviewer #2: All comments have been addressed

2. Is the manuscript technically sound, and do the data support the conclusions?

Reviewer #1: Yes

Reviewer #2: Yes

3. Has the statistical analysis been performed appropriately and rigorously? 

Reviewer #1: Yes

Reviewer #2: Yes

4. Have the authors made all data underlying the findings in their manuscript fully available?

Reviewer #1: Yes

Reviewer #2: Yes

5. Is the manuscript presented in an intelligible fashion and written in standard English?

Reviewer #1: Yes

Reviewer #2: Yes

6. Review Comments to the Author

Reviewer #1: 1. Look for PLOSONE policies regarding data and publication.

2. Please go thoroughly again regarding any typographical errors.

Thanks

Reviewer #2: The authors have adequately addressed my comments and made all data underlying the findings in their manuscript fully available. I think this study is acceptable for publishing in PLOS ONE.

7. PLOS authors have the option to publish the peer review history of their article (what does this mean?). If published, this will include your full peer review and any attached files.

Reviewer #1: No

Reviewer #2: No

---

## [Editor Report · Acceptance letter]

13 Jun 2022

PONE-D-21-40627R1 

Prevalence and associated risk factors for mental health problems among patients with polycystic ovary syndrome in Bangladesh: A nationwide cross- sectional study 

Dear Dr. Islam:

I'm pleased to inform you that your manuscript has been deemed suitable for publication in PLOS ONE. Congratulations! Your manuscript is now with our production department. 

Kind regards, 

on behalf of

Prof. Kenji Hashimoto 

Section Editor

PLOS ONE